Research

behaviour, palaeontology, evolution

Cretaceous, behaviour, egg sac,
Lagonomegopidae, maternal care, spiderling

**Authors for correspondence:**
Paul A. Selden
e-mail: selden@ku.edu
Dong Ren
e-mail: rendong@mail.cnu.edu.cn

# Maternal care in Mid-Cretaceous lagonomegopid spiders

Xiangbo Guo[1], Paul A. Selden[1,2,3] and Dong Ren[1]

[1]College of Life Sciences and Academy for Multidisciplinary Studies, Capital Normal University, 105 Xisanhuanbeilu, Haidian District, Beijing 100048, People's Republic of China
[2]Department of Geology, University of Kansas, 1414 Naismith Drive, Lawrence, KS 66045, USA
[3]Natural History Museum, Cromwell Road, London SW7 5BD, UK

XG, 0000-0002-7074-8642; PAS, 0000-0001-7454-4260; DR, 0000-0001-8660-0901

Maternal care benefits the survival and fitness of offspring, often at a cost to the mother's future reproduction, and has evolved repeatedly throughout the animal kingdom. In extant spider species, this behaviour is very common and has different levels and diverse forms. However, evidence of maternal care in fossil spiders is quite rare. In this study, we describe four Mid-Cretaceous (approx. 99 Ma) amber specimens from northern Myanmar with an adult female, part of an egg sac and some spiderlings of the extinct family Lagonomegopidae preserved, which suggest that adult lagonomegopid females probably built and then guarded egg sacs in their retreats or nests, and the hatched spiderlings may have stayed together with their mother for some time. The new fossils represent early evidence of maternal care in fossil spiders, and enhance our understanding of the evolution of this behaviour.

## 1. Introduction

Parental care refers to any investment by the parent that enhances the fitness of their offspring, and often at a cost to the survival and future reproduction of the parent [1,2]. Its evolution represents a breakthrough in the adaptation of animals to their environment and has significant implications for the evolution of sociality [3–5]. This behaviour has evolved independently numerous times among different animal groups [2] and has been observed in many arthropod groups including spiders. Fossil evidence of arthropod parental care has been reported in crustaceans [6,7], insects [8–10], whipspiders [11] and some early arthropods [12–14].

Spiders have a long geological history on Earth, from the Carboniferous period to the present [15,16]. In extant spider species, almost all known cases of parental care are maternal except *Manogea porracea*, an araneid spider that was reported with amphisexual care [5,17]. Maternal care in spiders is fairly common and has diverse forms [5,18]. However, fossils providing evidence of such behaviour are relatively rare. A number of fossil egg sacs have been found in Cenozoic (65.5 Ma to present) amber, including a sac carried in the chelicerae by a female synotaxid spider [19,20]. They represent the evidence of maternal care in Cenozoic fossil spiders. In addition, the earliest fossil records of spider egg sacs were reported from Mid-Cretaceous Burmese amber, but most of them are doubtful due to the lack of detailed description and photographs [21–23]. The early evolution of maternal care in spiders remains poorly known. Here, we describe an adult female, part of an egg sac and some spiderlings of Lagonomegopidae preserved in four pieces of Mid-Cretaceous (approx. 99 Ma) Burmese amber. The new fossils provide early evidence of maternal care in fossil spiders.

## 2. Material and methods

The amber specimens investigated in this paper are from Tanai Village in the Hukawng Valley, Myitkyina District of Kachin State, Myanmar. The amber-bearing

*Proc. R. Soc. B* **288**: 20211279

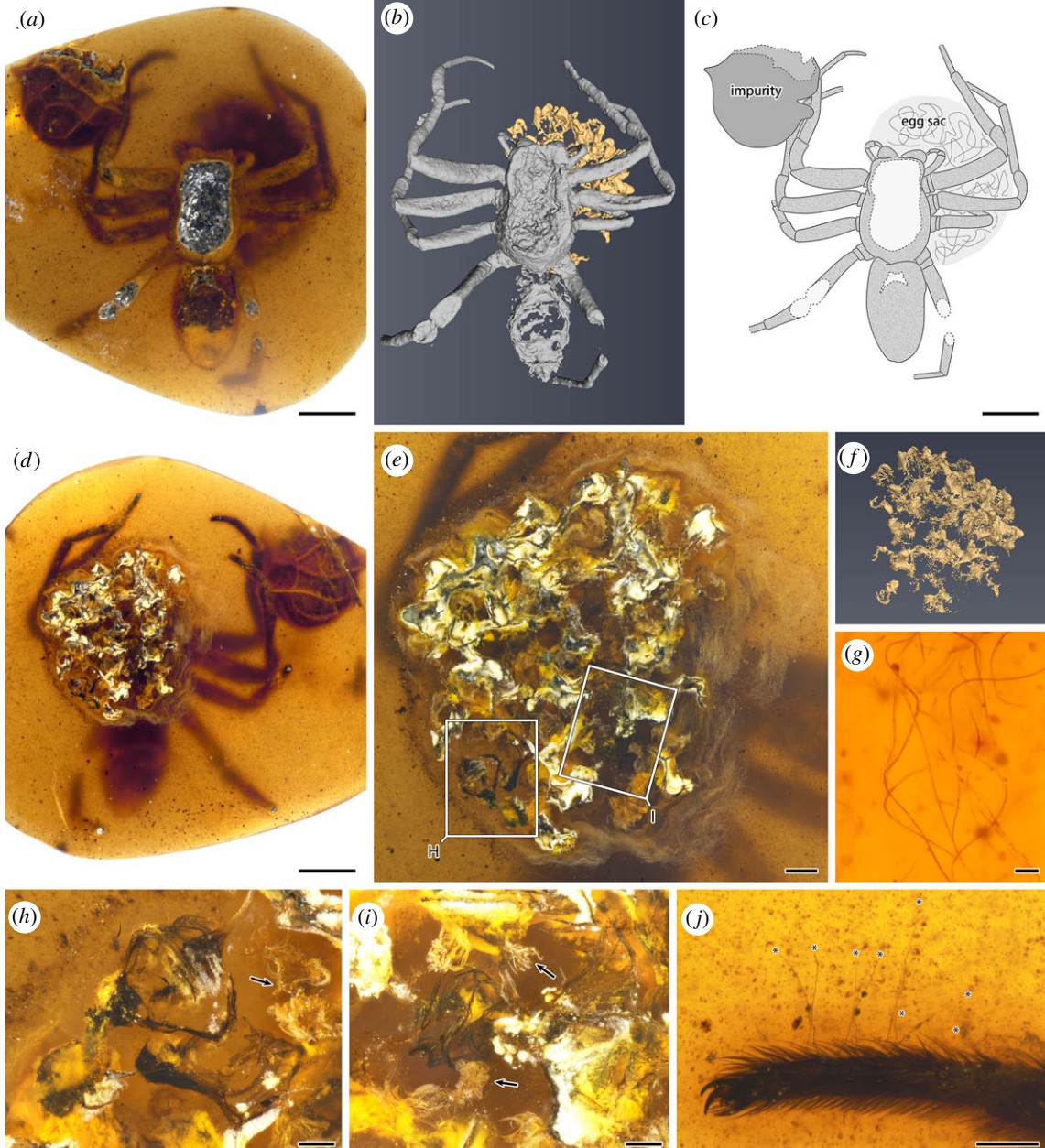

**Figure 1.** Photographs and drawings of a lagonomegopid spider (CNU-ARA-MA2016101) and egg sac in Burmese amber CNU009432. (*a*) Habitus, dorsal view; (*b*) three-dimensional CT reconstruction, dorsal view, grey represents CNU-ARA-MA2016101, yellow represents egg sac; (*c*) schematic drawing of dorsal habitus; (*d*) habitus, ventral view; (*e*) cross-section of egg sac, the details in boxes are enlarged and shown in other figures; (*f*) three-dimensional CT reconstruction of egg sac, dorsal view; (*g*) silk thread of the spider egg sac; (*h*) details of egg sac, showing prelarvae and egg membranes (arrow); (*i*) details of egg sac, showing prelarvae and egg membranes (arrows); (*j*) left tarsus I lateral view, showing trichobothria (asterisks). Scale bars represent 2 mm (*a,c,d*), 0.5 mm (*e*), 0.2 mm (*h,i,j*) and 0.01 mm (*g*); scale bars are absent in (*b*) and (*f*). (Online version in colour.)

deposits have been dated to the earliest Cenomanian, *ca* 98.8 ± 0.6 Ma, based on U–Pb radiometric dating of zircons from the volcaniclastic matrix [24]. All specimens are housed at the fossil collection of the Key Laboratory of Insect Evolution and Environmental Changes, at the College of Life Sciences, Capital Normal University, (CNUB; Dong Ren, curator), in Beijing, China.

Preparation and imaging methods follow Selden & Penney [25]. The photographs were taken with a Nikon SMZ 25 and an attached Nikon DS-Ri 2 digital camera system, as well as a Nikon ECLIPSE Ni and an attached Nikon DS-Ri 2 digital camera system. Micro-CT scanning of CNU009432 was carried out with a Micro-CT (MicroXCT-400, ZEISS), located at the Institute of Zoology, Chinese Academy of Sciences. The three-dimensional structure of CNU-ARA-MA2016101 and the egg sac was reconstructed using Avizo software. The line drawings were prepared with Adobe Illustrator CC and Adobe Photoshop CC, the images were processed by Adobe Photoshop CC.

## 3. Results

In this paper, four pieces of Burmese amber (CNU009371, CNU009431, CNU009432 and CNU009476) were studied. CNU009432 has a large spider (CNU-ARA-MA2016101) and part of an egg sac preserved (figure 1; electronic supplementary material, figure S1). The large spider is covered with emulsion-liked impurities, the dorsal parts of cephalothorax and abdomen are somewhat broken, and some of the leg podomeres are missing. Its large size, peg teeth on the promargin of the chelicera, unmodified pedipalps, spineless legs and trichobothria on the leg tarsus indicate that it belongs to Lagonomegopidae, and is an adult female [26]. Detailed description and measurements of CNU-ARA-MA2016101 can be found in electronic supplementary material, S1. The egg sac is broken and located under the female spider. From the cross section,

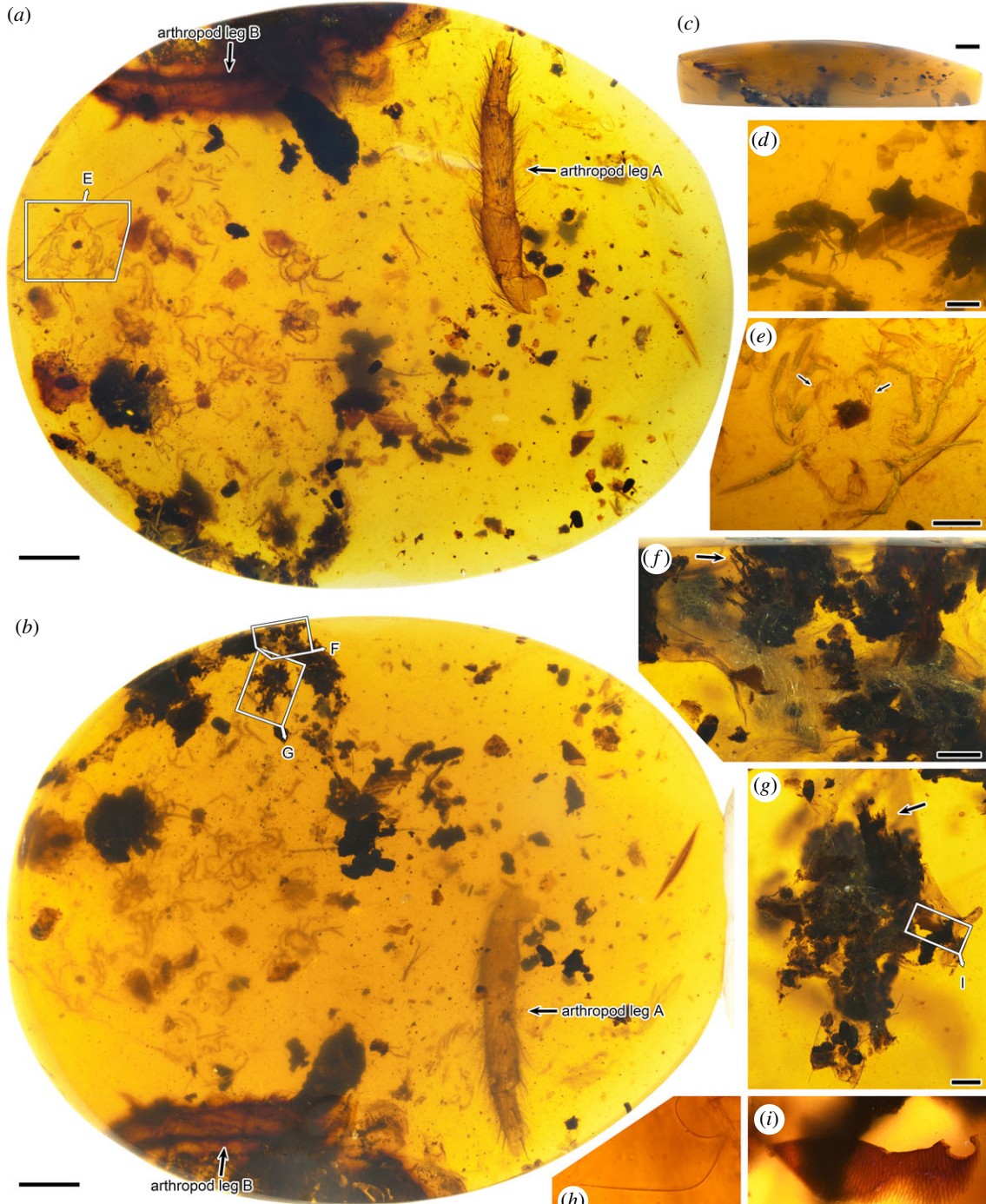

**Figure 2.** Photographs of inclusions in Burmese amber CNU009476. (*a*) Photograph of CNU009476, convex side view, the details in boxes are enlarged and shown in other figures; (*b*) photograph of CNU009476, flat side view, the details in boxes are enlarged and shown in other figures; (*c*) photograph of CNU009476, lateral view, showing the convex side and flat side; (*d*) a dipterous insect and several pieces of arthropod cuticular fragments; (*e*) overall habitus of CNU-ARA-MA2016123, dorsal view, showing the large PME situated on anterolateral corner of carapace (arrows); (*f*) spider silk threads and inclusions entwined by them, showing wood fibres (arrow); (*g*) inclusions entwined by spider silk, showing wood fibres (arrow), the details in box are enlarged and shown in other figure; (*h*) details of spider silk thread; (*i*) a piece of arthropod cuticular fragment. Scale bars represent 2 mm (*a,b,c*), 0.5 mm (*d,e*), 0.2 mm (*f,g*) and 0.025 mm (*h,i*). PME, posterior median eye(s). (Online version in colour.)

dozens of prelarvae and their egg membranes wrapped in the silk of the egg sac can be clearly observed (figure 1*e,g–i*). The diameter of the silk threads is about 1 µm.

CNU009476 has 24 spiderlings preserved, and is convex on one side and flat on the other in lateral view (figure 2; electronic supplementary material, figure S2). Most of the spiderlings are somewhat deformed and broken due to the effect of taphonomy; only semitransparent cuticles with setae and bristles remain. In addition, spider silk threads, some pieces

of arthropod cuticular fragments, several dipterous insects and parts of two large arthropod legs are preserved as syninclusions. The diameter of the silk threads is about 1 µm; they clump together loosely in a mass and entwine various pieces of detritus including wood fibres, arthropod cuticular fragments and some unidentifiable inclusions (figure 2*f–i*). CNU009431 has 26 spiderlings that are deformed due to the effect of taphonomy, and part of a large arthropod leg preserved; a cockroach is present as a syninclusion (figure 3*a–c*;

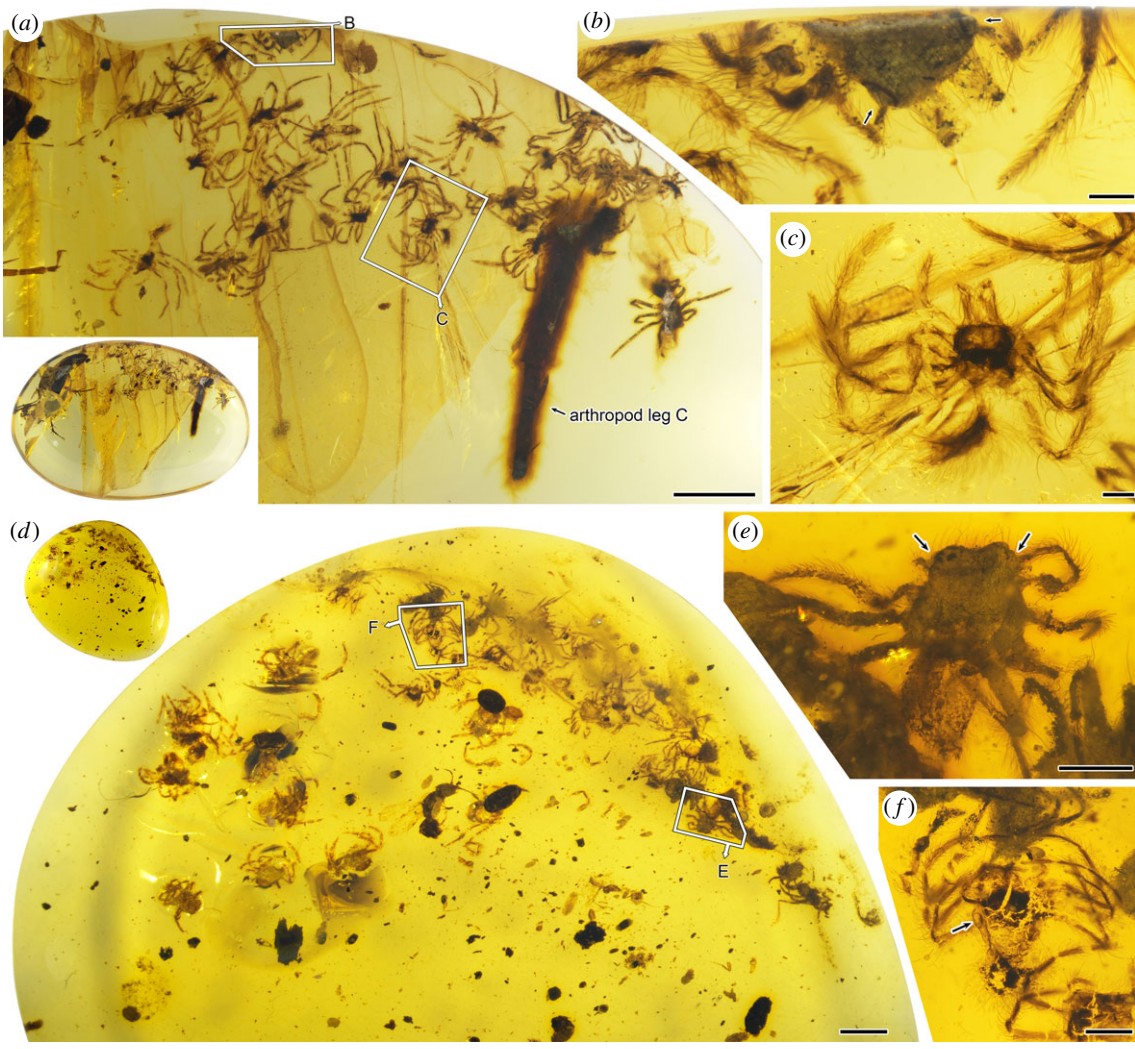

**Figure 3.** Photographs of inclusions in Burmese amber CNU009431 and CNU009371. (*a*) Photographs of CNU009431, overall photograph of amber on the bottom left, the details in boxes are enlarged and shown in other figures; (*b*) overall habitus of CNU-ARA-MA2016135, dorsal view, showing the large PME situated on anterolateral corner of carapace (arrows); (*c*) overall habitus of CNU-ARA-MA2016139, dorsal view; (*d*) photographs of CNU009371, overall photograph of amber on the top left, the details in boxes are enlarged and shown in other figures; (*e*) overall habitus of CNU-ARA-MA2016182, dorsal view, showing the large PME situated on anterolateral corner of carapace (arrows); (*f*) overall habitus of CNU-ARA-MA2016163, ventral view, showing the large PME situated on anterolateral corner of carapace (arrow). Scale bars represent 2 mm (*a,d*), 0.5 mm (*e,f*) and 0.2 mm (*b,c*). PME, posterior median eye(s). (Online version in colour.)

electronic supplementary material, figure S3). In CNU009371, a total of 34 spiderlings are preserved, most of them strongly taphonomically deformed (figure 3*d–f*; electronic supplementary material, figure S4). In addition, a wasp is present in CNU009371 as a syninclusion.

We consider that all spiderlings in the same piece of amber (CNU009371, CNU009431 and CNU009476) are siblings of the same instar because of their similar sizes and morphological characters. Their small sizes and immature morphological characters show that they were trapped by resin not long after hatching. Although most of the spiderlings are not well preserved, they can be identified as members of Lagonomegopidae by their two large eyes positioned on the anterolateral flanks of the carapace, peg teeth on the promargin of the chelicera, spineless legs and the presence of trichobothria on the leg tarsus. Detailed description and measurements of the lagonomegopid spiderlings in CNU009476 can be found in electronic supplementary material, data S2 (electronic supplementary material, figure S5).

In CNU009476, there are two parts of arthropod legs that have quite different characters preserved. One of them (figure 2*a,b*: arthropod leg A) has several macrosetae on the

surface of the podomeres, while the other (figure 2*a,b*: arthropod leg B) is spineless and has a kind of special lanceolate seta. Interestingly, such lanceolate setae can be observed on the part of the spineless arthropod leg preserved in CNU009431 (figure 3*a*: arthropod leg C), the leg of the large lagonomegopid spider preserved in CNU009432 and the type specimen of *Odontomegops titan* Guo *et al.* [26], which belongs to Lagonomegopidae as well (electronic supplementary material, figure S6).

## 4. Discussion

### (a) Maternal care in lagonomegopids

In extant spider species, the strategies of maternal care have different levels and diverse forms in different spider groups, ranging from laying the eggs in a sheltered place to feeding the young [5,18]. Spider eggs are always protected by silk, from a few silk threads wrapping around the eggs to having multiple inner layers of silk and a tough cocoon wall [18]. The spider egg sac is so common that its construction was not treated as maternal care in some previous studies [27]. In this study, we consider it as a form of maternal

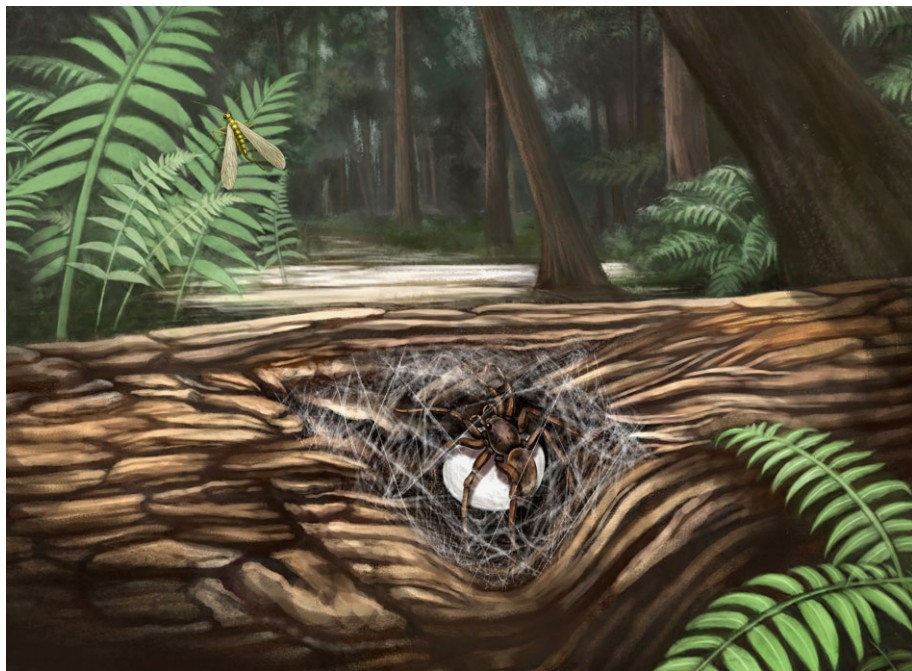

**Figure 4.** Ecological reconstruction of a female lagonomegopid spider guarding her egg sac. Painted by Xiaoran Zuo. (Online version in colour.)

care, under the definition proposed by Trivers [1] and Royle *et al.* [2]; thus, all spiders show a degree of maternal care [5]. The egg sac serves as a barrier against the entrance of parasites and predators [28–31]. The egg sac preserved in CNU009432 is composed of a loose mesh of threads and without special decorations, it represents an undoubted fossil record of a spider egg sac from Cretaceous amber.

In addition to protecting the eggs by building an egg sac, many modern spiders actively guard or carry their egg sacs. Spider egg sacs may be attached to the substrate and actively guarded by the adult female; others carry the egg sacs in their chelicerae (e.g. pholcids, scytodids, pisaurids), in their legs (e.g. huntsman spiders) or attached to the spinnerets (e.g. lycosids) until the young hatch (electronic supplementary material, figure S7). In CNU009432, the egg sac is preserved closely beneath the large spider. The eggs have developed into prelarvae, and some of them have shed their egg membranes (figure 1*e,h–i*). The prelarvae do not show enough characters that can be used to identify them to Lagonomegopidae or any other certain spider family. Nevertheless, the most reasonable interpretation of this piece of amber is still that the mother lagonomegopid spider was guarding her egg sac when trapped by resin. The maternal behaviour of egg guarding occurs in many spider families [5]. It plays an important role in protecting against egg predators [32,33] and regulating the temperature of egg sacs [34,35], thus benefitting the survival and successful hatching of eggs [5].

In CNU009476 and CNU009431, the spineless arthropod leg B and arthropod leg C have a kind of lanceolate seta which is also present in some lagonomegopid spiders, and the presence of spineless legs is a diagnostic feature of Lagonomegopidae. It hints that arthropod leg B and arthropod leg C possibly belong to two large lagonomegopid spiders, maybe the mothers of the spiderlings. In addition, unlike the silk of the egg sac in CNU009432, the loose silk threads in CNU009476 entwine various pieces of detritus, and are, therefore, probably part of a retreat or nest where the mother spider guarded her egg sac. This implies that the hatched lagonomegopid spiderlings may stay together with their mother in the retreat or nest for some time, rather than dispersing immediately.

In some living species, the mother spider provides food for the spiderlings by sharing prey [36], regurgitation [37], secreting nutritive liquid [38], trophic eggs [39] or matriphagy [5,18,40]. Although some pieces of arthropod cuticular fragments and several dipterous insects are preserved as syninclusions in CNU009476, there is no evidence indicating that offspring feeding behaviour existed or not in Lagonomegopidae. With the aforementioned evidence and discussions, we conclude that adult lagonomegopid females probably built and then guarded egg sacs in their retreats or nests, and the hatched spiderlings may stay together with their mother for some time (figure 4). It is worth mentioning that mother spiders with offspring fossilized in amber represent a moment in time, and the discoveries of more similar fossils, especially if the offspring are in different instars, will add material for further study of maternal care and even potential social behaviour in spiders.

## (b) Evolution of maternal care in spiders

Egg-sac building probably evolved with the origin of spiders. Egg guarding behaviour is relatively common in the major lineages of arachnids. Also, females of Amblypygi, Schizomida and Uropygi, which are suggested as the closest extant relatives of spiders in previous phylogenetic analyses [41–43], carry their eggs glued to their ventral abdomens until the young emerge, then carry the first instar offspring on their backs. Spiders probably evolved from egg guarding ancestors. The ability to produce silk is generally considered a defining feature of spiders [44,45]. One hypothesis for the origin of silk production in spiders is that silk evolved from proteinaceous secretions involved in construction of the egg sac in spider ancestors [46]. In other words, the original function of spider silk may have been to protect the eggs. Liphistiids, the sister group to all other extant spider species,

show some features that seem to be primitive for spiders [47]. They build globe-shaped egg sacs at the bottom of their burrows using silk and soil [48,49], but it is still unclear whether other maternal behaviours exist in this 'living fossil' family. CNU009432 shows the morphology of a spider egg sac and the putative egg-guarding behaviour from the Mid-Cretaceous, the fossil egg sac is similar to modern ones. Egg-sac building and guarding are probably conservative and successful strategies of maternal care in spiders.

More complex maternal care (offspring guarding and feeding) may have evolved independently multiple times across highly divergent spider lineages [5,50]. Even among closely related spider groups, the forms of maternal behaviour may be very different. For example, the egg sac of pirate spiders (Mimetidae) is usually laid on vegetation, then abandoned by the mother [47]. However, females of *Anansi insidiator* are known to carry the eggs and spiderlings with their chelicerae, and some species of *Mimetus* build nursery webs for the offspring [51]. Furthermore, some species of Theridiidae, which belongs to the same superfamily as Mimetidae [52,53], have a subsocial or even social life history [5,18].

Lagonomegopidae is an extinct spider family which was widespread in the Northern Hemisphere but only reported in the Cretaceous period [26,54]. A recent study suggested that it is the potential sister group to extant Palpimanoidea [55]. Our new fossils show that lagonomegopid females built an egg sac and may have guarded it, even possibly staying together with the hatched spiderlings for some time. In the five families of extant palpimanoid spiders, some archaeid and mecysmaucheniid females were observed and reported guarding their egg sacs [47,56,57], while maternal care in the other three palpimanoid families (Palpimanidae, Huttoniidae and Stenochilidae) is still unknown.

The evolution of maternal care is helpful for spiders in response to environmental pressures and represents an important step in the evolution of spider society [2,5]. The new fossils provide early evidence of maternal care (egg-sac building and guarding, and perhaps even offspring guarding) in fossil spiders, and enhance our understanding of the evolution of this behaviour.

**Ethics.** In this study, we reported 85 lagonomegopid specimens (CNU-ARA-MA2016101–2016185) preserved in four pieces of Burmese amber: CNU009371, CNU009431, CNU009432 and CNU009476. They are permanently housed in the Key Laboratory of Insect Evolution and Environmental Changes, College of Life Sciences and Academy for Multidisciplinary Studies, Capital Normal University, Beijing, China (CNUB; Dong Ren, Curator). All these amber specimens were collected from Kachin (Hukawng Valley) of northern Myanmar, and were dated at 98.79 ± 0.62 Ma based on U–Pb dating of zircons. They were acquired by Mr Fangyuan Xia before 2013 and donated for this study in 2015.

**Data accessibility.** The data supporting the conclusions of this article consist of (i) figures 1–3 and descriptions of the ambers included within the article; (ii) electronic supplementary material, figures S1–S7 and detailed descriptions of the lagonomegopid female and spiderlings which are provided in the electronic supplementary material; and (iii) the raw Micro-CT data and three-dimensional CT reconstruction of CNU009432, which are available from the Dryad Digital Repository: https://doi.org/10.5061/dryad.7m0cfxpv0 [58].

Supplementary information are provided in electronic supplementary material [59].

**Authors' contributions.** X.G.: Software, validation, visualization, writing-original draft; P.A.S.: supervision, validation, writing—original draft, writing—review and editing; D.R.: project administration, supervision, validation, writing—original draft, writing—review and editing.

All authors gave final approval for publication and agreed to be held accountable for the work performed therein.

**Competing interests.** We declare we have no competing interests.

**Funding.** This research is supported by grants from the National Natural Science Foundation of China (grant nos. 31730087 and 32020103006).

**Acknowledgements.** We thank Taiping Gao, Yongjie Wang, Lifang Xiao and Haoqiang Zhang of Capital Normal University for their helpful comments on this study. We are grateful to Caixia Gao (Institute of Zoology, Chinese Academy of Sciences) for providing help in reconstructing the three-dimensional structure of CNU-ARA-MA2016101. We thank Xiaoran Zuo for providing the ecological reconstruction picture. We thank Fangyuan Xia for donating specimens for this study. We thank the Editorial Board of *Proceedings B*, and in particular, Dr John Hutchinson. We express our gratitude to three anonymous reviewers for their valuable comments and suggestions.

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
