## [Peer Review File · Proceedings of the Royal Society B: Biological Sciences]

Review History

RSPB-2021-1279.R0 (Original submission)

Review form: Reviewer 1

Recommendation

Major revision is needed (please make suggestions in comments)

Scientific importance: Is the manuscript an original and important contribution to its field?

Good

General interest: Is the paper of sufficient general interest?

Good

Quality of the paper: Is the overall quality of the paper suitable?

Acceptable

Is the length of the paper justified?

No

Should the paper be seen by a specialist statistical reviewer?

No

Do you have any concerns about statistical analyses in this paper? If so, please specify them explicitly in your report.

No

It is a condition of publication that authors make their supporting data, code and materials available - either as supplementary material or hosted in an external repository. Please rate, if applicable, the supporting data on the following criteria.

Is it accessible?

Yes

Is it clear?

Yes

Is it adequate?

Yes

Do you have any ethical concerns with this paper?

No

Comments to the Author

In this manuscript the authors have shown a number of amber fossils with a female lagonomegopid spider with an egg sac, and other fossils with a small group of immature lagonomegopid spiderlings. It is relatively common in many spider families to remain with their egg sacs and, even, to have what Yip & Rayor (2013) termed 'transient maternal care' where the mother remains with their offspring for an instar or even two instars. The fossils appear to document transient maternal care in an extinct spider, but do not show any evidence of more prolonged associations between the spiderlings and their mother. This early example of maternal care in spiders it is rather exciting. However, I feel the authors have greatly oversold what their amber specimens demonstrate. Although it is reasonable to discuss the function of the spider egg sac, and range and complexity of maternal care and social behavior in spiders, the authors do not have the data to argue that adult females are doing anything beyond remaining with their egg sac and newly emerged offspring. Which is interesting in itself, but oversold. Similarly, many spider families build silk retreats or enclose themselves into a retreat composed of leaves, rocks, bark, that is sealed with silk where they lay their eggs (eg. sparassids). It is interesting that lagonomegopid spiders show evidence of a retreat which is enclosed with silk but can not be used to demonstrate extensive maternal care. The extent of the discussion should be reined in.

The authors repeatedly refer to the spiderlings as being 'deformed'. What does 'deformed' mean in this context? They were crushed by the sap? Appear to have birth defects? Are hard to assess in detail? What does it mean for the interpretation?

I have no experience interpreting details of specimens from amber, but is it possible that some of the young spiders were exuvia and not the spiderlings themselves? If only multiple molts were found, it might be used as an indication at what age the spiders disperse from their natal retreat.

My recommendation is to greatly shorten the Discussion. The discussion of spider egg sacs and potential predators is overlong. Similarly, it is worth mentioning the range of maternal and social behavior in spiders, but resist the urge to imply that these fossils demonstrate more than a brief period of maternal care until more fossils demonstrate more extensive associations.

Review form: Reviewer 2 (Jason Dunlop)

Recommendation

Accept with minor revision (please list in comments)

Scientific importance: Is the manuscript an original and important contribution to its field?

Good

General interest: Is the paper of sufficient general interest?

Good

Quality of the paper: Is the overall quality of the paper suitable?

Excellent

Is the length of the paper justified?

Yes

Should the paper be seen by a specialist statistical reviewer?

No

Do you have any concerns about statistical analyses in this paper? If so, please specify them explicitly in your report.

No

It is a condition of publication that authors make their supporting data, code and materials available - either as supplementary material or hosted in an external repository. Please rate, if applicable, the supporting data on the following criteria.

Is it accessible?

Yes

Is it clear?

Yes

Is it adequate?

Yes

Do you have any ethical concerns with this paper?

No

Comments to the Author

This is an interesting manuscript documenting remarkable examples of juvenile spiders associated with their egg sac and potentially also associated with their mother. Such examples of fossilised behavioural ecology are extremely rare in the amber record and merit inclusion in a high-profile, interdisciplinary journal. Figure 1 is especially convincing and impressive. However, the authors need to be a little careful in how they frame their results and shouldn't really claim that maternal care is PROVEN by their finds: it is a highly plausible hypothesis, but such behaviours can hardly ever be proven in fossils. They also need to cite any other putative examples of maternal care in the fossil spider literature in their Introduction.

The manuscript is well-written, except for a few linguistic problems, and nicely illustrated. The Discussion is a bit long and in places strays from the core message of spider maternal care. Overall, I would suggest MINOR REVISION and draw the authors attention to the points below.

ABSTRACT

Line 14: "...to adapt to their environment..."

Line 15: "However, evidence of..." [delete "the"]

Line 18: better "...of the extinct family Lagonomegopidae..."

Line 20: "...may have stayed together with..."

INTRODUCTION

Line 27: "...fitness of their offspring..."

Line 35: Given that the oldest spiders are Late Carboniferous, I'm not convinced they can really be considered part of the first wave of (arthropod) terrestrialisation, which took place about 100 million years earlier in the Silurian based on current data!

Line 38: "...behaviours are quite rare..." So are there previous examples in the literature? If so, then you need to cite them here! Off the top of my head some of the amber books by Jörg Wunderlich included extensive discussions of behavioural ecology in amber.

Line 40: I would use something like "putative evidence" instead of "certain evidence"; we can never be 100% sure with fossils that we are observing a given behaviour type - as opposed to a chance assemblage of (in this case) adult and juvenile fossils.

RESULTS

Line 74: better "...silk of an egg sac can be clearly observed..."

Line 85: better "...most of them are strongly deformed..."

Line 93: "...the presence of trichobothria..."

DISCUSSION

In general the discussion is quite long compared to the results. Any shortening here would be welcome.

Lines 98-99: "In living spiders, the strategies of maternal care have different levels and diverse forms in different spider groups, ranging from..."

Line 102: "...its construction was not treated..."

Line 104: better "...all spiders show a degree of maternal care..."

Line 107: delete the second citation "[28,30]", seems unnecessary twice in the same sentence.

Line 123: "...on the surface of the podomeres..."

Line 124: "...part of the arthropod leg..."

Lines 131-132: I wonder if its worth commenting on how the mother and young may have become preserved if they were hiding in a nest/retreat? Intuitively, I would expect amber to preferentially trap animals running around, especially on tree trunks, so how did something hiding in its retreat get caught? Alternatively what is the chance that lagonomegopids carried their egg sac with them like some living spiders? In Figure 1 does the CT scan reveal whether the chelicerae are embedded in (or at least in contact with) the egg sac?

Line 141: "...probably evolved with the..."

Line 145: "...spider silk may have been to protect..."

Line 160: "...an extinct spider family which was widespread..."

Lines 162-63: "...females built an egg sac and may have guarded it..."

Line 163: "...even probably cared for..."

Line 171: "Although spiders were probably..."

Line 172: "...predators during the Cretaceous, there were some..." [delete "Earth"]

Line 174: "...or preyed upon by certain..."

Line 175-180: How many of these predatory taxa are really relevant to the present discussion? Acroceridae, for example, attack adult spiders rather than their eggs as far as I recall, and thus maternal care is irrelevant here. I would concentrate here only on those predators which demonstrably attack spider eggs.

Line 181: "...eggs in the Cretaceous may also have been preyed on by ants..."

Line 188: "...and perhaps even potential offspring" [again, you can't be absolutely certain about this]

Line 190: Given that maternal care occurs sporadically even among, and within, living families, I'm not sure its presence in one (extinct) lineage tells us a huge amount about when these behaviours first arose in spiders in general. Be careful not to over-interpret the results.

REFERENCES

Line 240: "...450 million year-old..." [space missing]

Line 268: "Atlas of plants and animals..." [space missing]

Line 282: "Green lynx spider egg sacs: sources of..." [: missing]

Line 286: "Nørgarrd" not "Norgarrd" in the original citation.

Line 334: italicise "*gracilicollis*"

Line 368: should it be "...mid-Cretaceous..." [or is the mistake in the original citation?]

FIGURES / FIGURE LEGENDS

Line 379: "...drawing of a lagonomegopid spider (CNU..."

Line 396: "...by spider silk..." (or "silk threads")

Line 411: Why is it important to include the figure of the lanceolate setae? To show that the two new legs are probably lagonomegopids too? Maybe this should be briefly explained in the figure legend.

Line 418: "...of the new fossils..."

Is figure 5 entirely necessary? It doesn't bring a huge amount to the manuscript apart from showing the probable position of Lagonomegopidae. Its not detailed enough to, for example, map

examples of maternal care is other spider families.

Review form: Reviewer 3

Recommendation

Accept with minor revision (please list in comments)

Scientific importance: Is the manuscript an original and important contribution to its field?

Good

General interest: Is the paper of sufficient general interest?

Marginal

Quality of the paper: Is the overall quality of the paper suitable?

Acceptable

Is the length of the paper justified?

Yes

Should the paper be seen by a specialist statistical reviewer?

No

Do you have any concerns about statistical analyses in this paper? If so, please specify them explicitly in your report.

No

It is a condition of publication that authors make their supporting data, code and materials available - either as supplementary material or hosted in an external repository. Please rate, if applicable, the supporting data on the following criteria.

Is it accessible?

Yes

Is it clear?

Yes

Is it adequate?

Yes

Do you have any ethical concerns with this paper?

No

Comments to the Author

The authors describe four pieces of Burmese amber containing spiderlings of the extinct family Lagonomegopidae. One piece has an adult female spider over an egg sac, and two other species have leg fragments consistent with larger lagonomegopid spiders. Guarding egg sacs and the first instar out of the sac is extremely common in spiders, so it is expected that these behaviors are ancient, as well. While unsurprising, as the first fossil evidence for guarding behavior, the data presented in this paper are important for understanding the evolution of parental care. I found the paper straightforward and easy to read. However, in addition to minor comments and suggestions below, I also thought that parental care might be placed in a broader phylogenetic context. The closest relatives of spiders (and arachnids in general) guard their eggs and the first instar post-hatching. While egg sac construction evolved with spiders (as no other extant

arachnid order makes silken egg sacs like spiders), spiders probably evolved from egg-guarding ancestors. This behavior was then lost in some spider lineages. This isn't clear in the current manuscript. The phrase "positive response" in regards to maternal care evolution might imply that spiders evolved egg guarding from non-guarding ancestors, but that does not seem likely.

Specific comments:

Line 14: "Children" strongly implies personhood. I would suggest a more general term, like "offspring."

Line 14: I am unclear on what is meant by a "positive response." Is the idea here that maternal care is an adaptive behavior and not a product of drift or other non-selective processes? If so, I'm not certain that is necessary to mention.

Lines 14-15: I suggest changing the wording to "In extant spider species..." to make it clear you are talking about "living" at the species level, not the individual.

Line 21: As line 102 states that egg sac production is also considered parental care, I would think that fossil egg sacs with or without the mother would also be evidence of maternal care. I also thought the importance of study would be clearer if there were a statement along the lines of "X number of fossil egg sacs have been described, but none of them have also included fossil evidence for guarding females."

Line 28: I suggest specifying "future reproduction."

Line 30: Beneficial to what exactly? All adaptive behaviors are beneficial to the organism performing them (at least at the inclusive fitness level).

Line 36: Since that review was published there has been some evidence that males guard egg sacs in *Manogeta porracea* (Moura et al. *Animal Behav* 2017). The care seems to be fairly passive, but males hang around the female's web, even after the female is gone, and this increases the survivorship of eggs.

Line 71: It seems like there should be a reference here to *Lagonomegopidae* and its defining characteristics in the literature.

Line 109: Because the literature for fossil egg sacs is small, it might be worth providing more information here. What is the oldest spider egg sac? How does this one compare?

Lines 121-126: I suggest putting much of this information into the results. While there is some interpretation here, the results currently do not make clear that the leg fragments are important specimens for the question at hand. It would be helpful to inform the reader that these fragments are important in the results.

Line 124: Can the authors provide some idea as to how common these lanceolate setae are in arthropods? How confidently can they be ascribed to *Lagonomegopidae*, rather another arthropod group?

Line 126: Should that be 9432, rather than 9431, as this seems to refer to the adult female specimen?

Line 126: It would be helpful to specify that *Odontomegops titan* is a *lagonomegopid*.

Line 140: The results state that all spiderling groups were probably trapped in the resin not long after hatching. It might be worth mentioning that this fits the common pattern that spiderlings aggregate for an instar outside of the sac, during which time, they do not feed.

Line 171: Change "are" to "were" because the sentence refers to the Cretaceous.

Decision letter (RSPB-2021-1279.R0)

01-Jul-2021

Dear Dr Guo:

Your manuscript has now been peer reviewed and the reviews have been assessed by an Associate Editor. The reviewers' comments (not including confidential comments to the Editor) and the comments from the Associate Editor are included at the end of this email for your

reference. As you will see, the reviewers and the Editors have raised some concerns with your manuscript and we would like to invite you to revise your manuscript to address them.

2 things must be done w/ revised MS: 1) better phylogenetic context (polarity of maternal care evolution) as reviewer suggests, and 2) sharing of raw uCT data and ideally full 3D CT reconstructions (the figured specimens or any more), following our data sharing policy, e.g. in Dryad repository. The reviewer's point about the uncertainty of maternal care vs. other interpretations must be dealt with more clearly.

Research ethics:

Use of animals and field studies:

It is a condition of publication that you make available the data and research materials supporting the results in the article. Please see our Data Sharing Policies (<https://royalsociety.org/journals/authors/author-guidelines/#data>). Datasets should be deposited in an appropriate publicly available repository and details of the associated accession number, link or DOI to the datasets must be included in the Data Accessibility section of the article (<https://royalsociety.org/journals/ethics-policies/data-sharing-mining/>). Reference(s) to datasets should also be included in the reference list of the article with DOIs (where available).

If you wish to submit your data to Dryad (<http://datadryad.org/>) and have not already done so you can submit your data via this link [http://datadryad.org/submit?journalID=RSPB&manu=\(Document not available\)](http://datadryad.org/submit?journalID=RSPB&manu=(Document%20not%20available)), which will take you to your unique entry in the Dryad repository.

Please submit a copy of your revised paper within three weeks. If we do not hear from you within this time your manuscript will be rejected. If you are unable to meet this deadline please let us know as soon as possible, as we may be able to grant a short extension.

Best wishes,
Dr John Hutchinson
mailto: proceedingsb@royalsociety.org

Associate Editor
Board Member: 1
Comments to Author:
Dear Authors

You will see that all referees found the study of interest and have made numerous suggestions to improve clarity and have made some queries as well.

Referee 1 is concerned that the implications of these findings is perhaps a bit overstated. Would it be worthwhile making the conclusions more cautious? This is echoed by both reviewer 2 and 3 as well.

The manuscript is otherwise well received and with some revision to dampen the more far reaching conclusions that these findings evidence should make this a valuable paper.

Reviewer(s)' Comments to Author:
Referee: 1

Comments to the Author(s)

In this manuscript the authors have shown a number of amber fossils with a female lagonomegopid spider with an egg sac, and other fossils with a small group of immature lagonomegopid spiderlings. It is relatively common in many spider families to remain with their

egg sacs and, even, to have what Yip & Rayer (2013) termed 'transient maternal care' where the mother remains with their offspring for an instar or even two instars. The fossils appear to document transient maternal care in an extinct spider, but do not show any evidence of more prolonged associations between the spiderlings and their mother. This early example of maternal care in spiders is rather exciting. However, I feel the authors have greatly oversold what their amber specimens demonstrate. Although it is reasonable to discuss the function of the spider egg sac, and range and complexity of maternal care and social behavior in spiders, the authors do not have the data to argue that adult females are doing anything beyond remaining with their egg sac and newly emerged offspring. Which is interesting in itself, but oversold. Similarly, many spider families build silk retreats or enclose themselves into a retreat composed of leaves, rocks, bark, that is sealed with silk where they lay their eggs (eg. sparassids). It is interesting that lagonomegopid spiders show evidence of a retreat which is enclosed with silk but can not be used to demonstrate extensive maternal care. The extent of the discussion should be reined in.

The authors repeatedly refer to the spiderlings as being 'deformed'. What does 'deformed' mean in this context? They were crushed by the sap? Appear to have birth defects? Are hard to assess in detail? What does it mean for the interpretation?

I have no experience interpreting details of specimens from amber, but is it possible that some of the young spiders were exuvia and not the spiderlings themselves? If only multiple molts were found, it might be used as an indication at what age the spiders disperse from their natal retreat.

My recommendation is to greatly shorten the Discussion. The discussion of spider egg sacs and potential predators is overlong. Similarly, it is worth mentioning the range of maternal and social behavior in spiders, but resist the urge to imply that these fossils demonstrate more than a brief period of maternal care until more fossils demonstrate more extensive associations.

Referee: 2

Comments to the Author(s)

This is an interesting manuscript documenting remarkable examples of juvenile spiders associated with their egg sac and potentially also associated with their mother. Such examples of fossilised behavioural ecology are extremely rare in the amber record and merit inclusion in a high-profile, interdisciplinary journal. Figure 1 is especially convincing and impressive. However, the authors need to be a little careful in how they frame their results and shouldn't really claim that maternal care is PROVEN by their finds: it is a highly plausible hypothesis, but such behaviours can hardly ever be proven in fossils. They also need to cite any other putative examples of maternal care in the fossil spider literature in their Introduction.

The manuscript is well-written, except for a few linguistic problems, and nicely illustrated. The Discussion is a bit long and in places strays from the core message of spider maternal care. Overall, I would suggest MINOR REVISION and draw the authors attention to the points below.

ABSTRACT

Line 14: "...to adapt to their environment..."

Line 15: "However, evidence of..." [delete "the"]

Line 18: better "...of the extinct family Lagonomegopidae..."

Line 20: "...may have stayed together with..."

INTRODUCTION

Line 27: "...fitness of their offspring..."

Line 35: Given that the oldest spiders are Late Carboniferous, I'm not convinced they can really be considered part of the first wave of (arthropod) terrestrialisation, which took place about 100 million years earlier in the Silurian based on current data!

Line 38: "...behaviours are quite rare..." So are there previous examples in the literature? If so, then you need to cite them here! Off the top of my head some of the amber books by Jörg Wunderlich included extensive discussions of behavioural ecology in amber.

Line 40: I would use something like "putative evidence" instead of "certain evidence"; we can never be 100% sure with fossils that we are observing a given behaviour type - as opposed to a chance assemblage of (in this case) adult and juvenile fossils.

RESULTS

Line 74: better "...silk of an egg sac can be clearly observed..."

Line 85: better "...most of them are strongly deformed..."

Line 93: "...the presence of trichobothria..."

DISCUSSION

In general the discussion is quite long compared to the results. Any shortening here would be welcome.

Lines 98-99: "In living spiders, the strategies of maternal care have different levels and diverse forms in different spider groups, ranging from..."

Line 102: "...its construction was not treated..."

Line 104: better "...all spiders show a degree of maternal care..."

Line 107: delete the second citation "[28,30]", seems unnecessary twice in the same sentence.

Line 123: "...on the surface of the podomeres..."

Line 124: "...part of the arthropod leg..."

Lines 131-132: I wonder if its worth commenting on how the mother and young may have become preserved if they were hiding in a nest/retreat? Intuitively, I would expect amber to preferentially trap animals running around, especially on tree trunks, so how did something hiding in its retreat get caught? Alternatively what is the chance that lagonomegopids carried their egg sac with them like some living spiders? In Figure 1 does the CT scan reveal whether the chelicerae are embedded in (or at least in contact with) the egg sac?

Line 141: "...probably evolved with the..."

Line 145: "...spider silk may have been to protect..."

Line 160: "...an extinct spider family which was widespread..."

Lines 162-63: "...females built an egg sac and may have guarded it..."

Line 163: "...even probably cared for..."

Line 171: "Although spiders were probably..."

Line 172: "...predators during the Cretaceous, there were some..." [delete "Earth"]

Line 174: "...or preyed upon by certain..."

Line 175-180: How many of these predatory taxa are really relevant to the present discussion? Acroceridae, for example, attack adult spiders rather than their eggs as far as I recall, and thus maternal care is irrelevant here. I would concentrate here only on those predators which demonstrably attack spider eggs.

Line 181: "...eggs in the Cretaceous may also have been preyed on by ants..."

Line 188: "...and perhaps even potential offspring" [again, you can't be absolutely certain about this]

Line 190: Given that maternal care occurs sporadically even among, and within, living families, I'm not sure its presence in one (extinct) lineage tells us a huge amount about when these behaviours first arose in spiders in general. Be careful not to over-interpret the results.

REFERENCES

Line 240: "...450 million year-old..." [space missing]

Line 268: "Atlas of plants and animals..." [space missing]

Line 282: "Green lynx spider egg sacs: sources of..." [: missing]

Line 286: "Nørgarrd" not "Norgarrd" in the original citation.

Line 334: italicise "*gracilicollis*"

Line 368: should it be "...mid-Cretaceous..." [or is the mistake in the original citation?]

FIGURES / FIGURE LEGENDS

Line 379: "...drawing of a lagonomegopid spider (CNU..."

Line 396: "...by spider silk..." (or "silk threads")

Line 411: Why is it important to include the figure of the lanceolate setae? To show that the two new legs are probably lagonomegopids too? Maybe this should be briefly explained in the figure legend.

Line 418: "...of the new fossils..."

Is figure 5 entirely necessary? It doesn't bring a huge amount to the manuscript apart from showing the probable position of Lagonomegopidae. Its not detailed enough to, for example, map examples of maternal care in other spider families.

Referee: 3

Comments to the Author(s)

The authors describe four pieces of Burmese amber containing spiderlings of the extinct family Lagonomegopidae. One piece has an adult female spider over an egg sac, and two other species have leg fragments consistent with larger lagonomegopid spiders. Guarding egg sacs and the first instar out of the sac is extremely common in spiders, so it is expected that these behaviors are ancient, as well. While unsurprising, as the first fossil evidence for guarding behavior, the data presented in this paper are important for understanding the evolution of parental care. I found the paper straightforward and easy to read. However, in addition to minor comments and suggestions below, I also thought that parental care might be placed in a broader phylogenetic context. The closest relatives of spiders (and arachnids in general) guard their eggs and the first instar post-hatching. While egg sac construction evolved with spiders (as no other extant arachnid order makes silken egg sacs like spiders), spiders probably evolved from egg-guarding ancestors. This behavior was then lost in some spider lineages. This isn't clear in the current manuscript. The phrase "positive response" in regards to maternal care evolution might imply that spiders evolved egg guarding from non-guarding ancestors, but that does not seem likely.

Specific comments:

Line 14: "Children" strongly implies personhood. I would suggest a more general term, like "offspring."

Line 14: I am unclear on what is meant by a "positive response." Is the idea here that maternal care is an adaptive behavior and not a product of drift or other non-selective processes? If so, I'm not certain that is necessary to mention.

Lines 14-15: I suggest changing the wording to "In extant spider species..." to make it clear you are talking about "living" at the species level, not the individual.

Line 21: As line 102 states that egg sac production is also considered parental care, I would think that fossil egg sacs with or without the mother would also be evidence of maternal care. I also thought the importance of study would be clearer if there were a statement along the lines of "X number of fossil egg sacs have been described, but none of them have also included fossil evidence for guarding females."

Line 28: I suggest specifying "future reproduction."

Line 30: Beneficial to what exactly? All adaptive behaviors are beneficial to the organism performing them (at least at the inclusive fitness level).

Line 36: Since that review was published there has been some evidence that males guard egg sacs in *Manogeta porracea* (Moura et al. *Animal Behav* 2017). The care seems to be fairly passive, but males hang around the female's web, even after the female is gone, and this increases the survivorship of eggs.

Line 71: It seems like there should be a reference here to Lagonomegopidae and its defining characteristics in the literature.

Line 109: Because the literature for fossil egg sacs is small, it might be worth providing more information here. What is the oldest spider egg sac? How does this one compare?

Lines 121-126: I suggest putting much of this information into the results. While there is some interpretation here, the results currently do not make clear that the leg fragments are important specimens for the question at hand. It would be helpful to inform the reader that these fragments are important in the results.

Line 124: Can the authors provide some idea as to how common these lanceolate setae are in arthropods? How confidently can they be ascribed to Lagonomegopidae, rather another arthropod group?

Line 126: Should that be 9432, rather than 9431, as this seems to refer to the adult female specimen?

Line 126: It would be helpful to specify that *Odontomegops titan* is a lagonomegopid.

Line 140: The results state that all spiderling groups were probably trapped in the resin not long after hatching. It might be worth mentioning that this fits the common pattern that spiderlings aggregate for an instar outside of the sac, during which time, they do not feed.

Line 171: Change "are" to "were" because the sentence refers to the Cretaceous.

Author's Response to Decision Letter for (RSPB-2021-1279.R0)

See Appendix A.

RSPB-2021-1279.R1 (Revision)

Review form: Reviewer 1

Recommendation

Accept as is

Scientific importance: Is the manuscript an original and important contribution to its field?

Good

General interest: Is the paper of sufficient general interest?

Excellent

Quality of the paper: Is the overall quality of the paper suitable?

Excellent

Is the length of the paper justified?

Yes

Should the paper be seen by a specialist statistical reviewer?

No

Do you have any concerns about statistical analyses in this paper? If so, please specify them explicitly in your report.

No

It is a condition of publication that authors make their supporting data, code and materials available - either as supplementary material or hosted in an external repository. Please rate, if applicable, the supporting data on the following criteria.

Is it accessible?

Yes

Is it clear?

Yes

Is it adequate?

Yes

Do you have any ethical concerns with this paper?

No

Comments to the Author

In my previous suggestions to the authors, I felt that they oversold their observations of maternal care in the fossilized spiders. Here, the discussion and descriptions of maternal care in the spiders is handled with more discretion and is stronger for it. If anything, I think it would be worth adding a sentence at Line 143 indicating that mothers with offspring fossilized in amber

represents a moment in time, and that further researchers who discover more lagonomegopid (and other spider) fossils should be on the watch for more females with young, especially if the offspring are significantly larger or have large exuvia than those in this study indicating that the spiderlings are remaining with their mother beyond their emergence from the sac. Older instars of these spiderlings would give insight into potential subsocial or more complex social behavior in lagonomegopids.

Minor suggestions:

Line 38: This sentence is confusing with a misspelling. I suggest changing it to: A number of fossil egg sacs have been found in Cenozoic amber, including a sac carried by a 40 female synotaxid spider [19,20]. How was the synotaxid sac carried? In the chelicerae, legs, or on the spinnerets?

Line 118 - These sentences could be stronger. Many modern spiders actively guard or carry their egg sacs to reduce risk of predation and parasitization (many references). Spider egg sacs may be attached to the substrate and actively guarded by the adult female, others carry the egg sacs in their chelicerae (e.g., pholcids, scytodids, pisaurids), in their legs (e.g. huntsman) or attached to the spinnerets (e.g., lycosids) until the young hatch (Supplementary figure S7).

Otherwise, the manuscript is quite strong.

Review form: Reviewer 2 (Jason Dunlop)

Recommendation

Accept as is

Scientific importance: Is the manuscript an original and important contribution to its field?

Good

General interest: Is the paper of sufficient general interest?

Good

Quality of the paper: Is the overall quality of the paper suitable?

Good

Is the length of the paper justified?

Yes

Should the paper be seen by a specialist statistical reviewer?

No

Do you have any concerns about statistical analyses in this paper? If so, please specify them explicitly in your report.

No

It is a condition of publication that authors make their supporting data, code and materials available - either as supplementary material or hosted in an external repository. Please rate, if applicable, the supporting data on the following criteria.

Is it accessible?

Yes

Is it clear?

Yes

Is it adequate?

Yes

Do you have any ethical concerns with this paper?

No

Comments to the Author

The authors have made considerable efforts to address the concerns of the reviewers and have produced a shorter, more compact and more focussed manuscript. The results are better presented as hypotheses rather than facts and less relevant parts of the discussion have now been deleted. I think the manuscript is now ready, and appropriate for, publication in Proc. R. Soc. B.

Review form: Reviewer 3

Recommendation

Accept with minor revision (please list in comments)

Scientific importance: Is the manuscript an original and important contribution to its field?

Good

General interest: Is the paper of sufficient general interest?

Acceptable

Quality of the paper: Is the overall quality of the paper suitable?

Good

Is the length of the paper justified?

Yes

Should the paper be seen by a specialist statistical reviewer?

No

Do you have any concerns about statistical analyses in this paper? If so, please specify them explicitly in your report.

No

It is a condition of publication that authors make their supporting data, code and materials available - either as supplementary material or hosted in an external repository. Please rate, if applicable, the supporting data on the following criteria.

Is it accessible?

Yes

Is it clear?

Yes

Is it adequate?

Yes

Do you have any ethical concerns with this paper?

No

Comments to the Author

I am satisfied with the revised discussion about whether spiders evolved from egg-guarding ancestors. I have a few other minor comments below.

Lines 13-14: I think it is useful to define maternal care, but defining it as “caretaking activity

provided by the mother" doesn't really add much, as all of that is implicit in the words "maternal" and "care." While I questioned the meaning of "positive response" in the previous version, the new wording "meaningful" isn't any clearer to me. I suggest rewording the first sentence to something along the lines of "Maternal care benefits the survival and fitness of offspring often at a cost to the mother's future reproduction and has evolved repeatedly throughout the animal kingdom."

Line 38: I suggest dropping the word "particular," as it seems to contradict the previous sentence explaining that parental care in spiders has "diverse forms."

Line 39: It might be helpful to give the time span of the Cenozoic, so that it is clear that this new fossil evidence of maternal care is older than what was known previously.

Line 115-116: How many egg sacs are older? What is the oldest known spider egg sac? Being more specific than "one of the oldest" would help put this study in context.

Line 132-134: My understanding is that lagonomegopid spiders probably lacked a capture web. It may be worth mentioning that the construction of a retreat or nest is consistent with this foraging strategy, as many snare-building species suspend their egg sacs in or around the web.

Line 146: I suggest restricting this statement to just egg sac construction, rather than "maternal care, at least..." I don't see evidence that other forms of care were novel behaviors that evolved with the origin of spiders.

Line 150: It is also worth noting that mothers in all these other arachnid orders carry the first instar offspring on their backs.

Line 177: Again, the meaning of "meaningful" here is unclear to me.

Decision letter (RSPB-2021-1279.R1)

23-Aug-2021

Dear Dr Guo

I am pleased to inform you that your manuscript RSPB-2021-1279.R1 entitled "Maternal care in mid-Cretaceous lagonomegopid spiders" has been accepted for publication in Proceedings B. Congratulations!!

The referee(s) have recommended publication, but also suggest some minor revisions to your manuscript. Therefore, I invite you to respond to the referee(s)' comments and revise your manuscript. Because the schedule for publication is very tight, it is a condition of publication that you submit the revised version of your manuscript within 7 days. If you do not think you will be able to meet this date please let us know.

Sincerely,
 Dr John Hutchinson
 Editor, Proceedings B
 mailto:proceedingsb@royalsociety.org

Associate Editor:

Comments to Author:

Apologies for the delayed response. The associate editor was on holiday. Now I am back and I am happy to say that the three referees are happy with your revisions except for some final suggestions to be made for improvement and clarity.

Many congrats on the paper.

Reviewer(s)' Comments to Author:

Referee: 2

Comments to the Author(s)

The authors have made considerable efforts to address the concerns of the reviewers and have produced a shorter, more compact and more focussed manuscript. The results are better presented as hypotheses rather than facts and less relevant parts of the discussion have now been deleted. I think the manuscript is now ready, and appropriate for, publication in Proc. R. Soc. B.

Referee: 3

Comments to the Author(s)

I am satisfied with the revised discussion about whether spiders evolved from egg-guarding ancestors. I have a few other minor comments below.

Lines 13-14: I think it is useful to define maternal care, but defining it as "caretaking activity provided by the mother" doesn't really add much, as all of that is implicit in the words "maternal" and "care." While I questioned the meaning of "positive response" in the previous version, the new wording "meaningful" isn't any clearer to me. I suggest rewording the first sentence to something along the lines of "Maternal care benefits the survival and fitness of offspring often at a cost to the mother's future reproduction and has evolved repeatedly throughout the animal kingdom."

Line 38: I suggest dropping the word "particular," as it seems to contradict the previous sentence explaining that parental care in spiders has "diverse forms."

Line 39: It might be helpful to give the time span of the Cenozoic, so that it is clear that this new fossil evidence of maternal care is older than what was known previously.

Line 115-116: How many egg sacs are older? What is the oldest known spider egg sac? Being more specific than "one of the oldest" would help put this study in context.

Line 132-134: My understanding is that lagonomegopid spiders probably lacked a capture web.

It may be worth mentioning that the construction of a retreat or nest is consistent with this foraging strategy, as many snare-building species suspend their egg sacs in or around the web.

Line 146: I suggest restricting this statement to just egg sac construction, rather than "maternal care, at least..." I don't see evidence that other forms of care were novel behaviors that evolved with the origin of spiders.

Line 150: It is also worth noting that mothers in all these other arachnid orders carry the first instar offspring on their backs.

Line 177: Again, the meaning of "meaningful" here is unclear to me.

Referee: 1

Comments to the Author(s)

In my previous suggestions to the authors, I felt that they oversold their observations of maternal care in the fossilized spiders. Here, the discussion and descriptions of maternal care in the spiders is handled with more discretion and is stronger for it. If anything, I think it would be worth adding a sentence at Line 143 indicating that mothers with offspring fossilized in amber represents a moment in time, and that further researchers who discover more lagonomegopid

(and other spider) fossils should be on the watch for more females with young, especially if the offspring are significantly larger or have large exuvia than those in this study indicating that the spiderlings are remaining with their mother beyond their emergence from the sac. Older instars of these spiderlings would give insight into potential subsocial or more complex social behavior in lagonomegopids.

Minor suggestions:

Line 38: This sentence is confusing with a misspelling. I suggest changing it to: A number of fossil egg sacs have been found in Cenozoic amber, including a sac carried by a 40 female synotaxid spider [19,20]. How was the synotaxid sac carried? In the chelicerae, legs, or on the spinnerets?

Line 118 - These sentences could be stronger. Many modern spiders actively guard or carry their egg sacs to reduce risk of predation and parasitization (many references). Spider egg sacs may be attached to the substrate and actively guarded by the adult female, others carry the egg sacs in their chelicerae (e.g., pholcids, scytodids, pisaurids), in their legs (e.g. huntsman) or attached to the spinnerets (e.g., lycosids) until the young hatch (Supplementary figure S7).

Otherwise, the manuscript is quite strong.

Author's Response to Decision Letter for (RSPB-2021-1279.R1)

See Appendix B.

Decision letter (RSPB-2021-1279.R2)

25-Aug-2021

Dear Dr Guo

I am pleased to inform you that your manuscript entitled "Maternal care in mid-Cretaceous lagonomegopid spiders" has been accepted for publication in Proceedings B.

Your article has been estimated as being 9 pages long. Our Production Office will be able to confirm the exact length at proof stage.

Data Accessibility section

Open Access

Paper charges

Sincerely,

Appendix A

8 July, 2021

Dear Dr John Hutchinson,

Thank you for your kind letter and valuable comments and suggestions from Editors and three Reviewers on our manuscript “**Maternal care in mid-Cretaceous lagonomegopid spiders**”. After detailed consideration, we have accepted and adopted most comments, and revised our manuscript accordingly, especially the Discussion part in the main text. The revisions below are organized as responses to comments from the Editors and Reviewers in the e-mail.

Editors’ Comments

Comment 1. 2 things must be done w/ revised MS: 1) better phylogenetic context (polarity of maternal care evolution) as reviewer suggests, and 2) sharing of raw uCT data and ideally full 3D CT reconstructions (the figured specimens or any more), following our data sharing policy, e.g. in Dryad repository. The reviewer's point about the uncertainty of maternal care vs. other interpretations must be dealt with more clearly.

Authors’ Response. 1) We have added the discussion of spider maternal care in a broader phylogenetic context in the main text. 2) We have uploaded the raw Micro-CT data and 3D CT reconstruction of CNU009432 to Dryad repository (<https://doi.org/10.5061/dryad.7m0cfxpv0>), and added a Data accessibility statement in the manuscript. We have revised the corresponding statements in the main text. Thank you for the valuable comments.

Comment 2. Referee 1 is concerned that the implications of these findings is perhaps a bit overstated. Would it be worthwhile making the conclusions more cautious? This is echoed by both reviewer 2 and 3 as well.

Authors’ Response. We have changed the corresponding statements in the manuscript.

Reviewer 1’s Comments

Comment 1. However, I feel the authors have greatly oversold what their amber specimens demonstrate. Although it is reasonable to discuss the function of the spider egg sac, and range and complexity of maternal care and social behavior in spiders, the authors do not have the data to argue that adult females are doing anything beyond remaining with their egg sac and newly emerged offspring. Which is interesting in itself, but oversold. Similarly, many spider families build silk retreats or enclose themselves into a retreat composed of leaves, rocks, bark, that is sealed with silk where they lay their eggs (eg. sparassids). It is interesting that lagonomegopid spiders show evidence of a retreat which is enclosed with silk but can not be used to demonstrate extensive maternal care. The extent of the discussion should be reined in.

Authors' Response. We have changed the corresponding statements and shortened the Discussion in the manuscript. Thanks for the helpful comments.

Comment 2. The authors repeatedly refer to the spiderlings as being 'deformed'. What does 'deformed' mean in this context? They were crushed by the sap? Appear to have birth defects? Are hard to assess in detail? What does it mean for the interpretation?

Authors' Response. The spiderlings are deformed because of the effect of taphonomy. The deformation is taphonomic deformation. We changed the corresponding statements in the manuscript.

Comment 3. I have no experience interpreting details of specimens from amber, but is it possible that some of the young spiders were exuvia and not the spiderlings themselves? If only multiple molts were found, it might be used as an indication at what age the spiders disperse from their natal retreat.

Authors' Response. The young spiders were not exuvia. Some spiders in ambers have semitransparent body due to the effect of taphonomy, and may look like exuvia. While the exuvia of spiders usually have a strongly shrivelled abdomen which are not found in our specimens.

Comment 4. My recommendation is to greatly shorten the Discussion. The discussion of spider egg sacs and potential predators is overlong. Similarly, it is worth mentioning the range of maternal and social behavior in spiders, but resist the urge to imply that these fossils

demonstrate more than a brief period of maternal care until more fossils demonstrate more extensive associations.

Authors' Response. Thanks for your valuable recommendation. We have shortened the Discussion and changed the corresponding statements in the manuscript.

Reviewer 2's Comments

Comment 1. However, the authors need to be a little careful in how they frame their results and shouldn't really claim that maternal care is PROVEN by their finds: it is a highly plausible hypothesis, but such behaviours can hardly ever be proven in fossils. They also need to cite any other putative examples of maternal care in the fossil spider literature in their Introduction.

Authors' Response. Thanks for your valuable suggestions. We have changed the corresponding statements in the main text, and added other putative examples of maternal care in the fossil spider literature in the Introduction.

Comment 2. The Discussion is a bit long and in places strays from the core message of spider maternal care.

Authors' Response. We have shortened and revised the Discussion. Thanks for the suggestion.

Comment 3.

ABSTRACT

Line 14: "...to adapt to their environment..."

Line 15: "However, evidence of..." [delete "the"]

Line 18: better "...of the extinct family Lagonomegopidae..."

Line 20: "...may have stayed together with..."

INTRODUCTION

Line 27: "...fitness of their offspring..."

Authors' Response. We have corrected the corresponding statements in the main text.

Comment 4. Line 35: Given that the oldest spiders are Late Carboniferous, I'm not convinced they can really be considered part of the first wave of (arthropod) terrestrialisation, which took place about 100 million years earlier in the Silurian based on current data!

Authors' Response. We have corrected the corresponding statements and reference in the main text.

Comment 5. Line 38: "...behaviours are quite rare..." So are there previous examples in the literature? If so, then you need to cite them here! Off the top of my head some of the amber books by Jörg Wunderlich included extensive discussions of behavioural ecology in amber.

Authors' Response. We have added the corresponding statements and references in Introduction. Thanks for the suggestion.

Comment 6. Line 40: I would use something like "putative evidence" instead of "certain evidence"; we can never be 100% sure with fossils that we are observing a given behaviour type - as opposed to a chance assemblage of (in this case) adult and juvenile fossils.

Authors' Response. We have changed the corresponding statements in the main text. Thanks for the suggestion.

Comment 7.

RESULTS

Line 74: better "...silk of an egg sac can be clearly observed..."

Line 85: better "...most of them are strongly deformed..."

Line 93: "...the presence of trichobothria..."

DISCUSSION

Lines 98-99: "In living spiders, the strategies of maternal care have different levels and diverse forms in different spider groups, ranging from..."

Line 102: "...its construction was not treated..."

Line 104: better "...all spiders show a degree of maternal care..."

Line 107: delete the second citation "[28,30]", seems unnecessary twice in the same sentence.

Line 123: "...on the surface of the podomeres..."

Line 124: "...part of the arthropod leg..."

Authors' Response. We have corrected the corresponding statements in the main text.

Comment 8. Lines 131-132: I wonder if its worth commenting on how the mother and young may have became preserved if they were hiding in a nest/retreat? Intuitively, I would expect

amber to preferentially trap animals running around, especially on tree trunks, so how did something hiding in its retreat get caught? Alternatively what is the chance that lagonomegopids carried their egg sac with them like some living spiders? In Figure 1 does the CT scan reveal whether the chelicerae are embedded in (or at least in contact with) the egg sac?

Authors' Response. The CT scan shows that the chelicerae are not direct contact with the egg sac, they are separated from each other (Supplementary figure S1). There is no evidence indicating that lagonomegopids carried their egg sac with them. Though spiders' retreat/nest were usually built at covert places, they still have chance to be trapped by resin. Maybe that is a reason can explain why this kind of fossils are rare. It is difficult to infer the locations of lagonomegopid spiders' retreats/nests. But some of them may locate on the depression of tree trunks where can be trapped by resin relatively easily. Thanks for the comments.

Comment 9.

Line 141: "...probably evolved with the..."

Line 145: "...spider silk may have been to protect..."

Line 160: "...an extinct spider family which was widespread..."

Lines 162-63: "...females built an egg sac and may have guarded it..."

Line 163: "...even probably cared for..."

Line 171: "Although spiders were probably..."

Line 172: "...predators during the Cretaceous, there were some..." [delete "Earth"]

Line 174: "...or preyed upon by certain..."

Authors' Response. We have corrected the corresponding statements in the main text.

Comment 10. Line 175-180: How many of these predatory taxa are really relevant to the present discussion? Acroceridae, for example, attack adult spiders rather than their eggs as far as I recall, and thus maternal care is irrelevant here. I would concentrate here only on those predators which demonstrably attack spider eggs.

Authors' Response. We have removed this part from our manuscript. Thanks for the comments.

Comment 11.

Line 181: "...eggs in the Cretaceous may also have been preyed on by ants..."

Line 188: "...and perhaps even potential offspring" [again, you can't be absolutely certain about this]

Authors' Response. We have corrected the corresponding statements in the main text.

Comment 12. Line 190: Given that maternal care occurs sporadically even among, and within, living families, I'm not sure its presence in one (extinct) lineage tells us a huge amount about when these behaviours first arose in spiders in general. Be careful not to over-interpret the results.

Authors' Response. We have changed the corresponding statements in the main text.

Comment 13.

REFERENCES

Line 240: "...450 million year-old..." [space missing]

Authors' Response. There is a "-" between "million" and "year" in the original citation. We have corrected the corresponding statements in the main text.

Comment 14.

Line 268: "Atlas of plants and animals..." [space missing]

Line 282: "Green lynx spider egg sacs: sources of..." [: missing]

Line 286: "Nørgarrd" not "Norgarrd" in the original citation.

Authors' Response. We have corrected the corresponding statements in the main text.

Comment 15. Line 334: italicise "gracilicollis"

Authors' Response. The word "gracilicollis" is not italic in the original citation.

Comment 16.

Line 368: should it be "...mid-Cretaceous..." [or is the mistake in the original citation?]

FIGURES / FIGURE LEGENDS

Line 379: "...drawing of a lagonomegopid spider (CNU..." Line 396: "...by spider silk..." (or "silk threads")

Authors' Response. We have corrected the corresponding statements in the manuscript.

Comment 17. Line 411: Why is it important to include the figure of the lanceolate setae? To

show that the two new legs are probably lagonomegopids too? Maybe this should be briefly explained in the figure legend.

Authors' Response. This figure shows that arthropod leg B in CNU009476 and arthropod leg C in CNU009431 have a kind of lanceolate seta which also present in some lagonomegopid spiders. And these two arthropod legs are spineless, according with the diagnosis of Lagonomegopidae. It hints that arthropod leg B and arthropod leg C possibly belong to two large lagonomegopid spiders. We have moved this figure to Supplementary material as Supplementary figure S6. Thanks for the comments.

Comment 18.

Line 418: "...of the new fossils..."

Is figure 5 entirely necessary? It doesn't bring a huge amount to the manuscript apart from showing the probable position of Lagonomegopidae. Its not detailed enough to, for example, map examples of maternal care is other spider families.

Authors' Response. We have removed figure 5 and its legend from the manuscript.

Reviewer 3's Comments

Comment 1. However, in addition to minor comments and suggestions below, I also thought that parental care might be placed in a broader phylogenetic context. The closest relatives of spiders (and arachnids in general) guard their eggs and the first instar post-hatching. While egg sac construction evolved with spiders (as no other extant arachnid order makes silken egg sacs like spiders), spiders probably evolved from egg-guarding ancestors. This behavior was then lost in some spider lineages. This isn't clear in the current manuscript. The phrase "positive response" in regards to maternal care evolution might imply that spiders evolved egg guarding from non-guarding ancestors, but that does not seem likely.

Authors' Response. Thanks for your valuable recommendations on discussing maternal care in a broader phylogenetic context. We have added this part in Discussion and changed the corresponding statements in the manuscript.

Comment 2. Line 14: "Children" strongly implies personhood. I would suggest a more general

term, like “offspring.”

Authors’ Response. We have corrected the corresponding statements in the main text.

Comment 3. Line 14: I am unclear on what is meant by a “positive response.” Is the idea here that maternal care is an adaptive behavior and not a product of drift or other non-selective processes? If so, I’m not certain that is necessary to mention.

Authors’ Response. We have changed the corresponding statements in the main text.

Comment 4. Lines 14-15: I suggest changing the wording to “In extant spider species...” to make it clear you are talking about “living” at the species level, not the individual.

Authors’ Response. We have corrected the corresponding statements in the main text.

Comment 5. Line 21: As line 102 states that egg sac production is also considered parental care, I would think that fossil egg sacs with or without the mother would also be evidence of maternal care. I also thought the importance of study would be clearer if there were a statement along the lines of “X number of fossil egg sacs have been described, but none of them have also included fossil evidence for guarding females.”

Authors’ Response. We have added previous examples of fossil spider egg sacs in the Introduction. Thanks for the comments.

Comment 6. Line 28: I suggest specifying “future reproduction.”

Authors’ Response. We have corrected the corresponding statements in the main text.

Comment 7. Line 30: Beneficial to what exactly? All adaptive behaviors are beneficial to the organism performing them (at least at the inclusive fitness level).

Authors’ Response. We have corrected the corresponding statements in the main text.

Comment 8. Line 36: Since that review was published there has been some evidence that males guard egg sacs in *Manogeta porracea* (Moura et al. Animal Behav 2017). The care seems to be fairly passive, but males hang around the female’s web, even after the female is gone, and this increases the survivorship of eggs.

Authors’ Response. We have added the corresponding statements and reference in the main text.

Comment 9. Line 71: It seems like there should be a reference here to Lagonomegopidae and

its defining characteristics in the literature.

Authors' Response. We have added the corresponding reference in the main text.

Comment 10. Line 109: Because the literature for fossil egg sacs is small, it might be worth providing more information here. What is the oldest spider egg sac? How does this one compare?

Authors' Response. We have moved this part to Introduction, and added the corresponding statements in the main text.

Comment 11. Lines 121-126: I suggest putting much of this information into the results. While there is some interpretation here, the results currently do not make clear that the leg fragments are important specimens for the question at hand. It would be helpful to inform the reader that these fragments are important in the results.

Authors' Response. We have moved this part to the RESULTS and changed the corresponding statements in the main text. Thanks for the suggestion.

Comment 12. Line 124: Can the authors provide some idea as to how common these lanceolate setae are in arthropods? How confidently can they be ascribed to Lagonomegopidae, rather than another arthropod group?

Authors' Response. There are many types of setae in arthropods especially spiders. But morphological studies on these virous setae are relatively scarce. Though we did not find any reports of these lanceolate setae on other arthropods, we cannot treat the presence of lanceolate setae on legs as a crucial character of Lagonomegopidae. So, we surmise that arthropod leg B and arthropod leg C possibly belong to two large lagonomegopid spiders based on not only the presence of lanceolate setae, but also the spineless surface. Thanks for the comments.

Comment 13. Line 126: Should that be 9432, rather than 9431, as this seems to refer to the adult female specimen?

Authors' Response. We have corrected the corresponding statements in the main text.

Comment 14. Line 126: It would be helpful to specify that *Odontomegops titan* is a lagonomegopid.

Authors' Response. We have added the corresponding statements in the main text.

Comment 15. Line 140: The results state that all spiderling groups were probably trapped in

the resin not long after hatching. It might be worth mentioning that this fits the common pattern that spiderlings aggregate for an instar outside of the sac, during which time, they do not feed.

Authors' Response. There is no evidence can help us conclude whether the spiderlings feed or not during the aggregate time. We'd better be more careful in framing our results. We have changed the corresponding statements in the main text.

Comment 16. Line 171: Change “are” to “were” because the sentence refers to the Cretaceous.

Authors' Response. We have corrected the corresponding statements in the main text.

We hope all these corrections and revisions will be satisfactory and this manuscript will be acceptable for publication in **Proceedings of the Royal Society B**.

Sincerely yours, and on behalf of our coauthors,

Dr. Dong Ren

College of Life Sciences and

Academy for Multidisciplinary Studies

Capital Normal University

Beijing, 100048, China

E-mail: rendong@cnu.edu.cn

Dr. Paul A. Selden

Department of Geology

University of Kansas

Lawrence KS 66045, USA.

E-mail: selden@ku.edu

Appendix B

25 August, 2021

Dear Dr John Hutchinson,

Thanks for the acceptance of our manuscript “**Maternal care in mid-Cretaceous lagonomegopid spiders**” for publication in Proceeding B. And we are grateful to Editors and three Reviewers for providing valuable comments and suggestions on our manuscript. After detailed consideration, we have accepted and adopted most comments, and revised our manuscript accordingly. Moreover, we added an ecological reconstruction picture in the manuscript to show the egg-sac guarding behaviour in lagonomegopid spiders (figure 4 in our manuscript). The revisions below are organized as responses to comments from the Reviewers in the e-mail.

Reviewer 1’s Comments

Comment 1. In my previous suggestions to the authors, I felt that they oversold their observations of maternal care in the fossilized spiders. Here, the discussion and descriptions of maternal care in the spiders is handled with more discretion and is stronger for it. If anything, I think it would be worth adding a sentence at Line 143 indicating that mothers with offspring fossilized in amber represents a moment in time, and that further researchers who discover more lagonomegopid (and other spider) fossils should be on the watch for more females with young, especially if the offspring are significantly larger or have large exuvia than those in this study indicating that the spiderlings are remaining with their mother beyond their emergence from the sac. Older instars of these spiderlings would give insight into potential subsocial or more complex social behavior in lagonomegopids.

Authors’ Response. We have added the corresponding statements in the manuscript. Thanks for the helpful comments.

Comment 2. Line 38: This sentence is confusing with a misspelling. I suggest changing it to: A number of fossil egg sacs have been found in Cenozoic amber, including a sac carried by a 40 female synotaxid spider [19,20]. How was the synotaxid sac carried? In the chelicerae, legs,

or on the spinnerets?

Authors' Response. The female synotaxid spider carried her sac in the chelicerae. We have changed the corresponding statements in the manuscript.

Comment 3. Line 118 - These sentences could be stronger. Many modern spiders actively guard or carry their egg sacs to reduce risk of predation and parasitization (many references). Spider egg sacs may be attached to the substrate and actively guarded by the adult female, others carry the egg sacs in their chelicerae (e.g., pholcids, scytodids, pisaurids), in their legs (e.g. huntsman) or attached to the spinnerets (e.g., lycosids) until the young hatch (Supplementary figure S7).

Authors' Response. We have changed the corresponding statements in the manuscript. Thanks for the comments.

Reviewer 3's Comments

Comment 1. Lines 13-14: I think it is useful to define maternal care, but defining it as “caretaking activity provided by the mother” doesn't really add much, as all of that is implicit in the words “maternal” and “care.” While I questioned the meaning of “positive response” in the previous version, the new wording “meaningful” isn't any clearer to me. I suggest rewording the first sentence to something along the lines of “Maternal care benefits the survival and fitness of offspring often at a cost to the mother's future reproduction and has evolved repeatedly throughout the animal kingdom.”

Authors' Response. Thanks for the helpful comments. We have changed the corresponding statements in the manuscript.

Comment 2. Line 38: I suggest dropping the word “particular,” as it seems to contradict the previous sentence explaining that parental care in spiders has “diverse forms.”

Authors' Response. We have changed the corresponding statements in the main text.

Comment 3. Line 39: It might be helpful to give the time span of the Cenozoic, so that it is clear that this new fossil evidence of maternal care is older than what was known previously.

Authors' Response. We have added the corresponding statements in the manuscript.

Comment 4. Line 115-116: How many egg sacs are older? What is the oldest known spider

egg sac? Being more specific than “one of the oldest” would help put this study in context.

Authors’ Response. The oldest fossil record of spider egg sac is also reported from mid-Cretaceous Burmese amber, the corresponding statements can be found in Introduction. We have changed this sentence to “it represents an undoubted fossil record of a spider egg sac from Cretaceous amber”. Thanks for the helpful comments.

Comment 5. Line 132-134: My understanding is that lagonomegopid spiders probably lacked a capture web. It may be worth mentioning that the construction of a retreat or nest is consistent with this foraging strategy, as many snare-building species suspend their egg sacs in or around the web.

Authors’ Response. Many modern spiders (e.g., some anyphaenids, clubionids, gnaphosids and salticids) lack a capture web but have a retreat or nest for resting, hiding from their enemies and laying eggs. So, we think the construction of a retreat or nest is probably independent from the foraging strategy in spiders.

Comment 6. Line 146: I suggest restricting this statement to just egg sac construction, rather than “maternal care, at least...” I don’t see evidence that other forms of care were novel behaviors that evolved with the origin of spiders.

Authors’ Response. We have changed the corresponding statements in the main text.

Comment 7. Line 150: It is also worth noting that mothers in all these other arachnid orders carry the first instar offspring on their backs.

Authors’ Response. We have added the corresponding statements in the main text.

Comment 8. Line 177: Again, the meaning of “meaningful” here is unclear to me.

Authors’ Response. We have changed the corresponding statements in the manuscript.

Thank you again for your great help on this manuscript.

Sincerely yours, and on behalf of our coauthors,

Dr. Dong Ren

College of Life Sciences and

Academy for Multidisciplinary Studies

Capital Normal University

Beijing, 100048, China

E-mail: rendong@cnu.edu.cn

Dr. Paul A. Selden

Department of Geology

University of Kansas

Lawrence KS 66045, USA.

E-mail: selden@ku.edu